# Rolling bearing fault diagnosis based on Gramian angular difference field and improved channel attention model



Lunpan Wei[1], Xiuyan Peng[1] and Yunpeng Cao[2]

[1] College of Intelligent Systems Science and Engineering, Harbin Engineering University, Harbin, Heilongjiang, China
[2] College of Power and Energy Engineering, Harbin Engineering University, Harbin, Heilongjiang, China

## ABSTRACT

Fault diagnosis of rolling bearings is a critical task, and in previous research, convolutional neural networks (CNN) have been used to process vibration signals and perform fault diagnosis. However, traditional CNN models have certain limitations in terms of accuracy. To improve accuracy, we propose a method that combines the Gramian angular difference field (GADF) with residual networks (ResNet) and embeds frequency channel attention module (Fca) in the ResNet to diagnose rolling bearing fault. Firstly, we used GADF to convert the signals into RGB three-channel fault images during data preprocessing. Secondly, to further enhance the performance of the model, on the foundation of the ResNet we embedded the frequency channel attention module with discrete cosine transform (DCT) to form Fca, to effectively explores the channel information of fault images and identifies the corresponding fault characteristics. Finally, the experiment validated that the accuracy of the new model reaches 99.3% and the accuracy reaches 98.6% even under an unbalanced data set, which significantly improves the accuracy of fault diagnosis and the generalization of the model.

## INTRODUCTION

Rolling bearings, as typical components of rotating machinery, play an indispensable role in practical equipment and production. Because they normally work under complex working conditions such as load variations, they are prone to faults, resulting in costly downtime and catastrophic consequences. To ensure the normal operation of mechanical equipment, real-time monitoring (*Zhang et al., 2019*) of vibration signals generated by rotating machinery is necessary.

CNN is a type of deep learning network that plays a crucial role in the field of fault diagnosis by utilizing its unique convolutional layers to extract distinct features. In the study conducted by *Chen, Zhang & Gao (2021)*, they employed a one-dimensional convolutional neural network (1D-CNN) and a long short-term memory network (LSTM) to process signals, and achieved excellent classification results through training and classification tasks. *Wang et al. (2020)* first de-noised the signal, then input the processed

Corresponding author
Yunpeng Cao,
caoyunpeng0908@163.com

signal into CNN for automatic feature extraction, and finally performed feature partitioning and classification. However, these methods have limitations in solely deepening the network layers for feature extraction, as they may suffer from degradation issues. To address this, the ResNet was introduced. *Kong & Wang (2021)* introduced the residual structure into the Inception network to form a parallel structure, to alleviate the degradation problem caused by network deepening, which fully demonstrated the performance of fault classification. *Che et al. (2021)* proposed a novel approach that utilizes a deep residual contraction network to handle multiple faults and long-term sequences of vibration signals in rolling bearing systems. This method demonstrates excellent robustness in dealing with noisy samples and maintains a high accuracy in fault diagnosis.

The emergence of attention mechanisms further enhances the feature extraction capability of residual networks. *Xie, Wang & Shi (2023)* proposed a diagnostic method that integrates multi-scale convolution and attention mechanisms, which can maintain a high diagnostic accuracy even under variable load and noise interference conditions. *Li, Wang & Xie (2023)* introduced the Convolutional Block Attention Module (CBAM) into residual networks, of which the result shows that this method has a simple structure, easy to implement, and can effectively extract features. However, such methods only utilize the original signals as inputs, which may lead to incomplete feature extraction. Therefore, *Zhu et al. (2022)* proposed a novel approach where they transformed a series of fault signals into two-dimensional gray-scale images using wavelet transform. And then they feed these images into a residual attention network for classification. This network can automatically learn and highlight the key features in image input, so as to improve the accuracy classification. On the other side, *Qiu, Tao & Cheng (2022)* employed a different approach. First, they transformed the vibration signals of rolling bearings into two-dimensional frequency domain feature gray-scale images using fast Fourier transform. Then they utilized an auxiliary classifier generative adversarial network (ACGAN) to classify these images. Research results indicate that using two-dimensional feature images as input is more suitable for the recognition task of CNN.

The fault diagnosis method proposed in this article fully utilizes the three-channel encoded images of the Gramian angular difference field (GADF) and designs a residual network model with an embedded frequency channel attention module. Combining the advantages of both, can fully exploit fault information and improve diagnostic accuracy. Firstly, the signal is transformed into a two-dimensional image by using GADF encoding. Secondly, it is fed into the Fca-ResNet model for fault classification. Finally, through validation with bearing data from the Western Reserve University, it shows that this method performs best, compared with different models with a sufficient amount of balanced samples. Even under imbalanced data samples, this model still demonstrates excellent generalization capabilities.

## GRAMIAN ANGULAR DIFFERENCE FIELD

GADF is an encoding method that can transform one-dimensional signals into RGB three-channel images (*Wang & Oates, 2015*). This encoding method generates three channels during the data preprocessing stage, aligning with the characteristic of Fca-ResNet's

emphasis on channel information. When performing fault diagnosis, we aim to extract more useful information. GADF, while generating two-dimensional images, also preserves temporal information. Therefore, we choose to use GADF as the encoding method.

Normalize the one-dimensional data by rearranging the measured values $X = \{x_1, x_2, ..., x_n\}$ within the interval $[-1, 1]$.

$$\bar{x}^i_{-1} = \frac{(x_i - \max(X) + (x_i - \min(X)))}{\max(X) - \min(X)} \tag{1}$$

The rearranged time series is denoted as $\bar{X}$. The numerical values are encoded as angular cosine and displayed in polar coordinates.

$$\begin{cases} \phi_i = \arccos(\bar{x}_i), & -1 \leq \bar{x}_i \leq 1, \bar{x}_i \in \bar{X} \\ r_i = \dfrac{t_i}{N}, & t_i \in N \end{cases} \tag{2}$$

The formula contains a constant $N$ used to regulate the span and $t_i$ represents the radius. In order to retain temporal information, the values corresponding to time are distributed at certain angles on the span circle. GADF constructs a new feature representation by calculating the angular differences between different time points in a time series. These angular differences can be used to represent the dynamic changes in the time series data. Mapping these angle differences onto a circle can enable time information to be encoded in a more intuitive way, as the circular structure can effectively represent periodic changes and trends. Consequently, even with reduced data dimensions, GADF can still preserve key dynamic features within the time series.

After converting to a polar coordinate system, taking into account the triangular differences between each point, the correlation within different time intervals is represented using the angle perspective method. Therefore, the definition of GADF is as follows.

$$\begin{aligned} \text{GADF} &= [\sin(\phi_i - \phi_j)] \\ &= \sqrt{I - \bar{X}'^2} \cdot \bar{X} - \bar{X}' \cdot \sqrt{I - \bar{X}^2} \end{aligned} \tag{3}$$

In the above equation, $I$ represents a unit row vector, while $\bar{X}$ and $\bar{X}'$ are two distinct row vectors.

Figure 1 (GADF encoding process) illustrates the encoding process of GADF. As we can see that the encoding follows the trend of the time-series signal, moving from the top left to the bottom right. This method allows the changes along the time axis to be naturally mapped onto the encoded representation, thereby facilitating subsequent data analysis and model training. Moreover, it can retain information in the temporal dimension. When the one-dimensional vibration signal fluctuates, the corresponding GADF encoding image will show prominent cross patterns, and the larger the fluctuation amplitude, the more obvious the cross is. Therefore, GADF images can reflect impulsive signals and capture the fault characteristics of the vibration process.

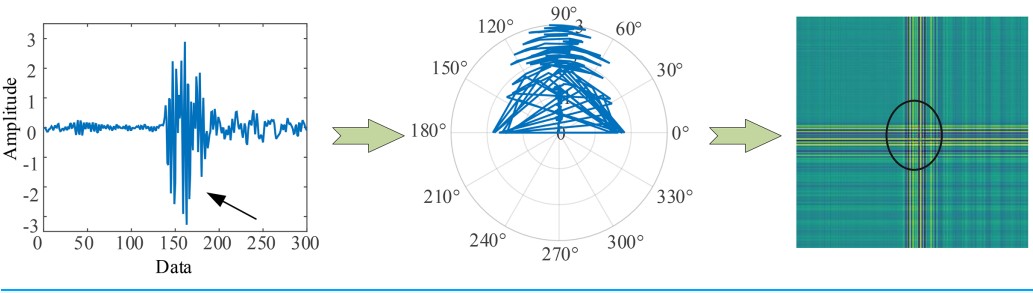

**Figure 1** GADF encoding process.               

## RESIDUAL STRUCTURE

The most commonly used method for traditional CNN models to achieve stronger feature extraction capabilities was to increase the depth of the model. However, when the model's depth reaches a certain level, further increasing the number of layers can lead to a decrease in accuracy, resulting in a phenomenon known as model degradation. The introduction of residual networks has largely addressed this problem (*He et al., 2016*). By using residual blocks, the depth of the model can reach over 1,000 layers without experiencing model degradation.

The basic principle of residual networks is to use a shortcut connection channel between convolutional layers, which allows the effective features extracted from the previous layer to be directly transmitted to the subsequent layers through this channel. This approach prevents the convolutional layers from redundantly extracting the features already captured by the previous layers. After passing through the shortcut channel $x$, the features obtained from the main channel $F(x)$ are fused with the features from the shortcut connection, to gain the output features $H(x)$ of the residual unit.

$$H(x) = F(x) + x \tag{4}$$

This structure effectively reduces the impact of poorly extracted features from the upper layers on the subsequent convolutional layers, thereby reducing the likelihood of the model getting trapped in local optima during training. The model structure is illustrated in Fig. 2 (Residual connections).

## FREQUENCY CHANNEL ATTENTION MODEL

The channel attention model enhances the accuracy of classification tasks by adaptively focusing on useful channel information. The introduction of this model enables neural networks to better comprehend and leverage crucial features within the input data, so as to improve performance across various application domains. *Qin, Zhang & Wu (2021)* utilized the DCT from the signal processing field and proposed a multi-spectral attention module. This method allows for better aggregation of frequency energy. And the principle of this method aligns with the fundamental concept of the channel attention model. Therefore, incorporating it into the channel attention model is a highly effective attempt. This article uses 2D DCT, and the working principle is briefly explained as below.

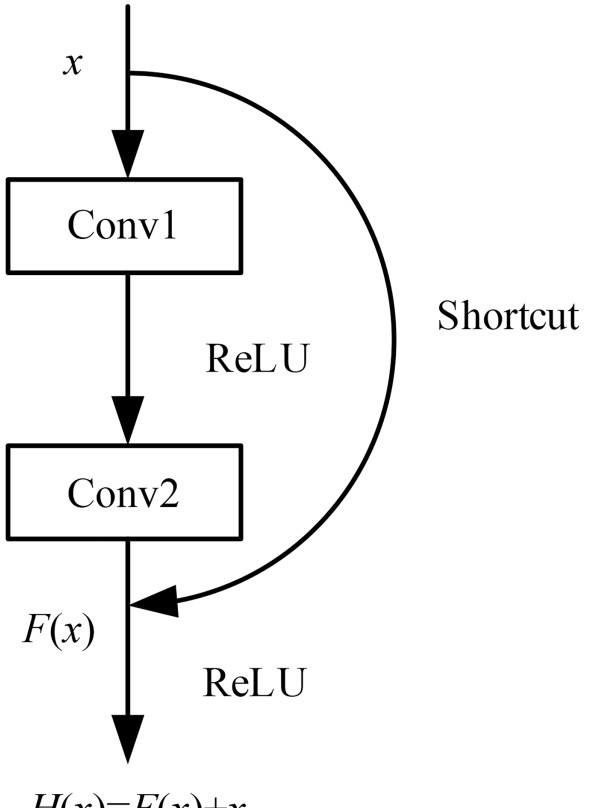

*x*

Conv1

ReLU

Conv2

*F(x)*

ReLU

*H(x)=F(x)+x*

**Figure 2 Residual connections.**

The basic principle of 2D-DCT is defined as follows.

$$f_{h,w}^{2d} = \sum_{i=0}^{H-1} \sum_{j=0}^{W-1} x_{i,j}^{2d} \cos\left(\frac{\pi h}{H}\left(i+\frac{1}{2}\right)\right) \cos\left(\frac{\pi w}{W}\left(j+\frac{1}{2}\right)\right) \tag{5}$$

In the equation, $f$ represents the spectrum of the DCT, $x$ denotes the input, $H$ refers to the height of the feature map, and $W$ represents the width of the feature map.

As depicted in Fig. 3 (Fca module), let $X$ represent the input feature map, and $C$ denote the number of channels. Divide the channels into $n$ groups, with each group containing $C'$ channels.

$$C' = \frac{C}{n} \tag{6}$$

In order to simplify the equation, the symbol $B$ is used to represent the basic form of the 2D-DCT in Eq. (6).

$$B_{h,w}^{i,j} = \cos\left(\frac{\pi h}{H}\left(i+\frac{1}{2}\right)\right) \cos\left(\frac{\pi w}{W}\left(j+\frac{1}{2}\right)\right) \tag{7}$$

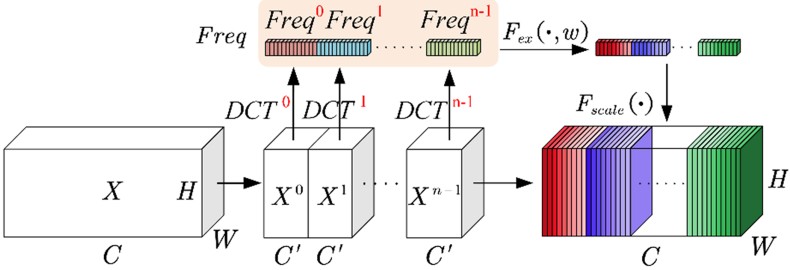

**Figure 3 Fca module.**

The parameter $[u, v]$ represents the allocated frequency components for each group, and these parameters are pre-defined. The basic form of the multi-spectral attention module is a $C'$ dimensional vector denoted as $Freq^i$.

$$Freq^i = \sum_{h=0}^{H-1} \sum_{w=0}^{W-1} X^i_{:,h,w} B^{u,v}_{h,w} \tag{8}$$

The overall pre-processing vector $Freq$ is formed by concatenating these basic vectors together.

$$Freq = cat\left(\left[Freq^0, Freq^1, \ldots, Freq^{n-1}\right]\right) \tag{9}$$

The above describes the squeezing operation. It is precisely due to the multi-spectral attention module enables Fca to obtain more comprehensive channel information during the squeezing phase.

During the excitation phase, in order to reduce computational complexity, compression is performed through a fully connected layer. After activation by the ReLU function, it is passed through another fully connected layer and activated by the Sigmoid function to obtain the $S$:

$$S = F_{ex}(Z, W) = \sigma(g(Z, W)) = \sigma(W_2 \delta(W_1 Z)) \tag{10}$$

In the equation, $F_{ex}$ represents the activation of the Sigmoid function, serving as the excitation operation. $W_1$ denotes the compression through the first fully connected layer. $\delta(\cdot)$ represents the activation by the ReLU function. Lastly, $W_2$ corresponds to the release through the second fully connected layer.

The operation $F_{scale}(\cdot)$ in Fig. 3 multiplies the reassembled $S$ with the original feature matrix groups, to obtain a new set of feature matrix groups.

## CONSTRUCTION OF FCA-RESNET MODEL ARCHITECTURE

The base model used in this article is a 34-layer ResNet. Frequency channel attention modules are embedded in the convolutional blocks of the intermediate layers, forming the Fca-ResNet model. The input images are processed into $224 \times 224$ images in the preprocessing stage. Therefore, in order to obtain a larger receptive field, the first convolutional layer uses a convolutional kernel with the size of $7 \times 7$. The convolutional

**Table 1 Fca-ResNet parameter settings.**

| Layout | Feature dimension | Hierarchy design |
|---|---|---|
| Input | $224 \times 224 \times 3$ | —— |
| Convolutional layer | $112 \times 112 \times 64$ | $7 \times 7 \times 64$, $s = 2$ |
| Maximum pooling layer | $56 \times 56 \times 64$ | $3 \times 3$, $s = 2$ |
| Fca $\times 1$ | $56 \times 56 \times 64$ | $3 \times 3 \times 64 \times 3$ $3 \times 3 \times 64 \times 3$ |
| Fca $\times 2$ | $28 \times 28 \times 128$ | $3 \times 3 \times 128 \times 4$ $3 \times 3 \times 128 \times 4$ |
| Fca $\times 3$ | $14 \times 14 \times 256$ | $3 \times 3 \times 256 \times 6$ $3 \times 3 \times 256 \times 6$ |
| Fca $\times 4$ | $7 \times 7 \times 512$ | $3 \times 3 \times 512 \times 3$ $3 \times 3 \times 512 \times 3$ |
| ReLU | $7 \times 7 \times 512$ | —— |
| Average pooling layer | $1 \times 1 \times 512$ | $7 \times 7$, $s = 1$ |
| Fully connected layer | $1 \times 1 \times 1{,}000$ | —— |
| Softmax | 10 | —— |

kernels in the intermediate layers are set to the size of $3 \times 3$. After average pooling, the input is passed to fully connected layers for classification, and the results of classification diagnosis are obtained through the Softmax layer. The specific design of model parameters is shown in Table 1. The overall structure of the model is illustrated in Fig. 4.

## EXPERIMENTAL RESULTS AND ANALYSIS

### Data preprocessing

The data used in the experiment was obtained from the Case Western Reserve University (CWRU) (*Case Western Reserve University, 2013*). Taking the drive-end rolling bearing SKF-6205 as an example, researchers created single-point damages of 0.007, 0.014, and 0.021 in (1 in = 25.4 mm) by electric discharge machining at the locations of inner ring, outer ring (at 3, 6, and 12 o'clock positions), and rolling element respectively.

To validate the superiority of the Fca-ResNet model, the experiment divided the load of the motor into four categories (0, 1, 2, 3 hp), while this article selected the condition of 1 hp (approximately 0.735 kw) and a speed of 1,772 r/min; And selected three different fault diameters of three types, namely outer ring (@6, where @6 indicates the fault location at 6 o'clock), inner ring, and rolling element as data samples, as well as one sample of normal state data, 10 categories in total. The number of sampling points of a cycle of the bearing is 400, and 300 data points were selected each time (the points collected within 3/4 cycles) (*Tong, Pang & Wei, 2021*). The original vibration signals were processed using a sliding window approach with a step size of 150. Using the overlapping sampling method, 400 samples were constructed for each category. These samples were then divided into training and validation sets in accordance with a specific ratio, as shown in Table 2. The specific model training parameters are detailed in Table 3. The transformed fault graphs are illustrated in Fig. 5.

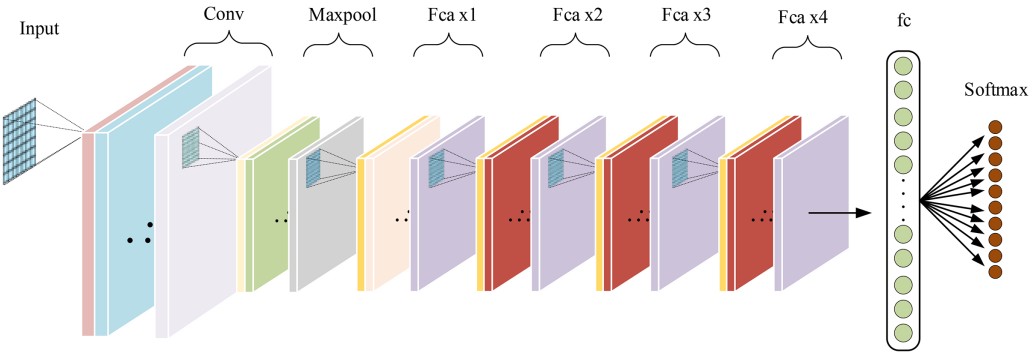

**Figure 4** Fca-ResNet model architecture.     

**Table 2  Fault samples.**

| Fault type | Label | Fault diameter/in | Training set | Validation set |
|---|---|---|---|---|
| Normal | 0 | – | 360 | 40 |
| Inner ring failure | 1 | 0.007 | 360 | 40 |
| Rolling element failure | 2 | 0.007 | 360 | 40 |
| Outer ring failure (@6) | 3 | 0.007 | 360 | 40 |
| Inner ring failure | 4 | 0.014 | 360 | 40 |
| Rolling element failure | 5 | 0.014 | 360 | 40 |
| Outer ring failure (@6) | 6 | 0.014 | 360 | 40 |
| Inner ring failure | 7 | 0.021 | 360 | 40 |
| Rolling element failure | 8 | 0.021 | 360 | 40 |
| Outer ring failure (@6) | 9 | 0.021 | 360 | 40 |
| Total | – | – | 3,600 | 400 |

**Table 3  Model training parameters.**

| Batch size | Loss function | Optimizer | Learning rate | Epoch |
|---|---|---|---|---|
| 32 | CrossEntropy loss | Adam | 0.001 | 300 |

## Analysis of experimental results

To validate the reliability of our model, we conducted experiments using a partitioned dataset. The comparative experimental models selected were ResNet, ResNet with SE attention modules, group-processed ResNeXt, and traditional CNN models such as GoogleNet and AlexNet. Figure 6 clearly demonstrates the validation accuracy and training loss variations of different models during the training process. As shown in the graph, ResNet, SE-ResNet, and ResNeXt achieved the highest validation accuracies of 98.5%, 98.8%, 98.5%, 98%, and 96.5% respectively. The fluctuation stability was average and the accuracy was not high. While, after 50 rounds of training, Fca-ResNet model

**Peer**J Computer Science

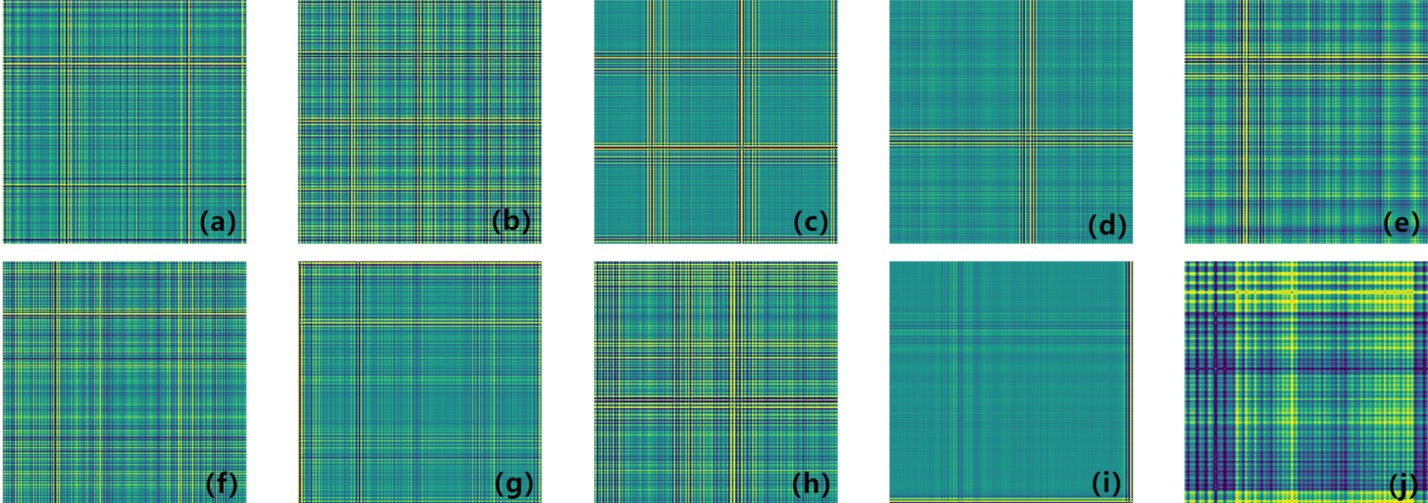

**Figure 5 (A–J) Coding diagram of different fault types.** (A) 0.007 in inner ring failure; (B) 0.014 in inner ring failure; (C) 0.021 in inner ring failure; (D) 0.007 in rolling element failure; (E) 0.014 in rolling element failure; (F) 0.021 in rolling element failure; (G) 0.007 in outer ring failure; (H) 0.014 in outer ring failure; (I) 0.021 in outer ring failure; (J) normal.

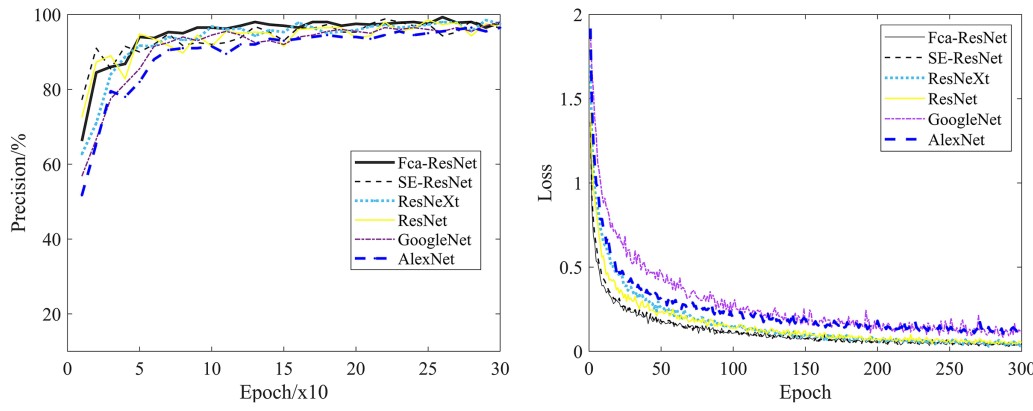

**Figure 6 Validation accuracy and training loss of different models.**

tended to stabilize, reaching a peak validation accuracy of 99.3%, and exhibited the lowest training loss compared to other models in the first 100 rounds. This indicates that our model possesses strong robustness and feature extraction capabilities, far superior to the other models considered.

In Fig. 7, we compared the classification results of multiple models on the validation set. By observing the confusion matrix, we can gain a visual understanding of how the models perform in different categories. Each node on the horizontal and vertical axes of the confusion matrix represents a different fault type, and the values on the diagonal indicate the degree of correspondence between the correct labels and predicted labels. From Fig. 7A, it can be observed that Fca-ResNet only misclassified three images, that is, two labels of the 4th class (0.014 in inner ring failure) were misclassified as the 5th class (0.014 in rolling element failure) and the 6th class (0.014 in outer ring failure (@6)) labels; and

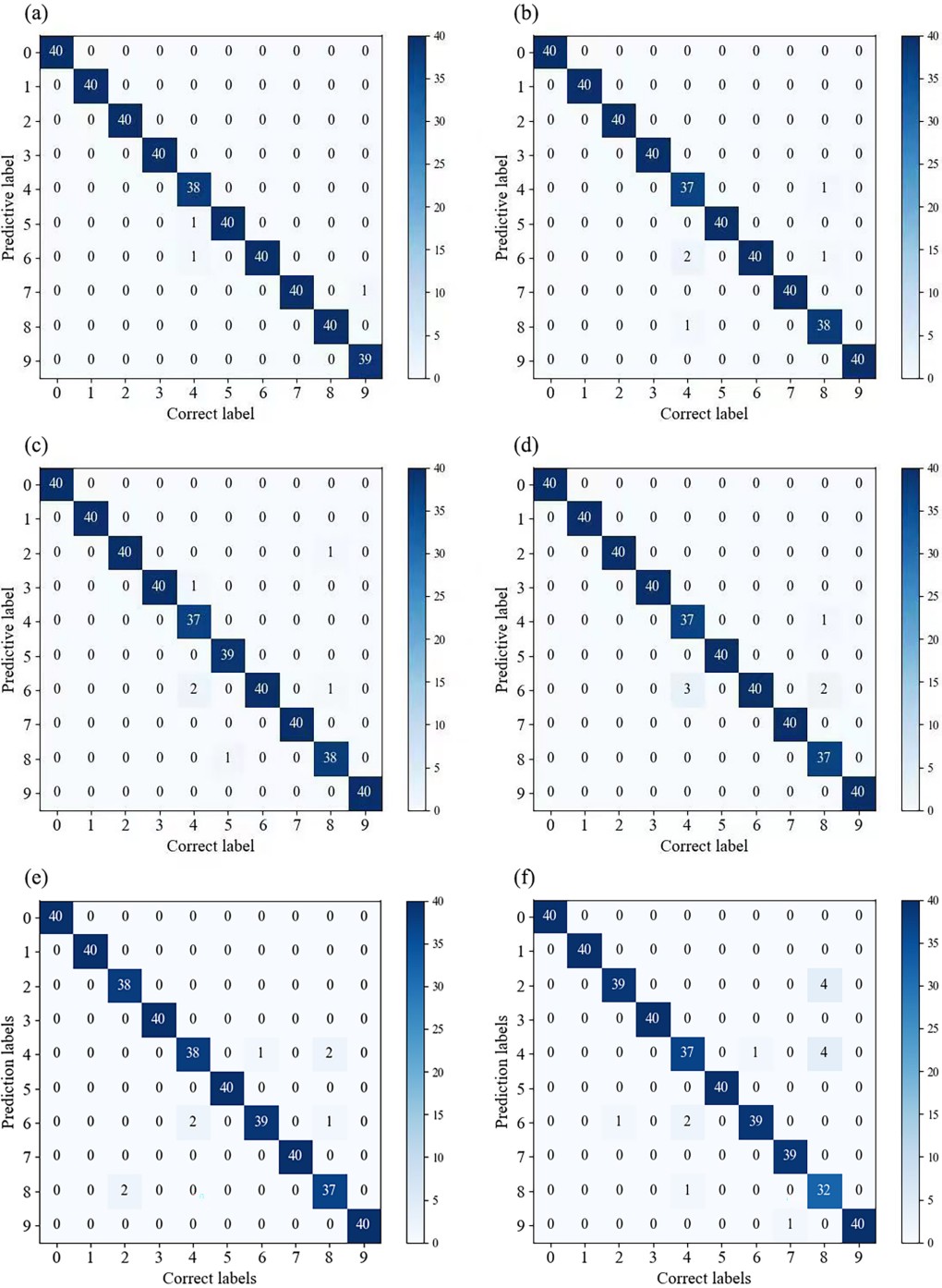

**Figure 7 (A–F) Confusion matrices for different models' classification.** (A) Confusion matrix for Fca-ResNet classification; (B) confusion matrix for SE-ResNet classification; (C) confusion matrix for ResNeXt classification; (D) confusion matrix for ResNet classification; (E) confusion matrix for GoogleNet classification; (F) confusion matrix for AlexNet classification.

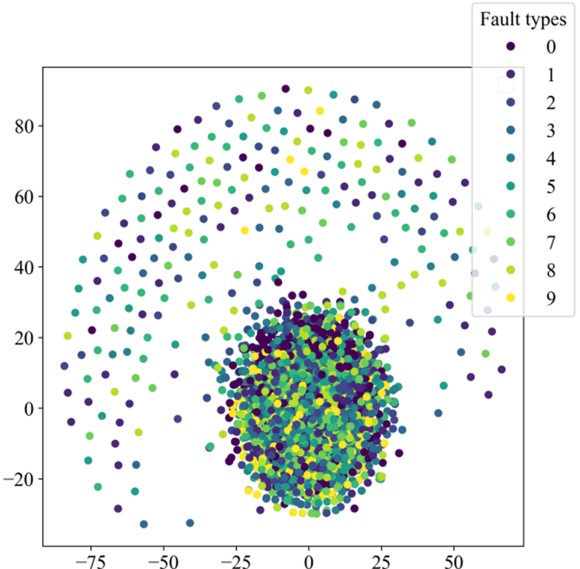

**Figure 8 Distribution of high-dimensional feature information in the original dataset.**

one label of the 9th class (0.014 in outer ring failure (@6)) was misclassified as the 7th class (0.021 in inner ring failure) label. However, all other data achieved a perfect classification result. Although several other models achieved full classification for multiple fault types, their individual performance for certain faults was inferior to that of this model.

## Visualization experiment results of different models

This section will validate the effectiveness of various model architectures for different features' extraction, using the t-distributed stochastic neighbor embedding (t-SNE) method to visualize the training set in the dataset. The main purpose of t-SNE is to reduce the feature space. Similar categories are modeled by nearby points, dissimilar categories are modeled by high-probability distant points, to simplify a high-dimensional dataset into a low-dimensional feature map that retains a large amount of original information, and clustering to visualize the distribution of different features.

Figure 8 illustrates the distribution of high-dimensional feature information in the original dataset, while Fig. 9 presents the visual clustering effects of the fully connected layers in different models. From Fig. 8, it can be observed that the high-dimensional feature information of the original dataset is scattered throughout the sample space, and each type of feature information is randomly mixed together. Figure 9A shows that the clustering effect of the last fully connected layer in Fca-ResNet almost completely separates the data samples of different fault types, and directly clustered samples of the same type. Only a few categories are incorrectly clustered into other categories. The visualization of features aligns perfectly with the confusion matrix. From Figs. 9B–9F, it can be observed that SE-ResNet and ResNeXt exhibit significant overlap between two fault classes, while ResNet, GoogleNet, and AlexNet demonstrate substantial overlap among several fault classes. Though all these models achieve the accuracy of over 96.5%, their dimensionality

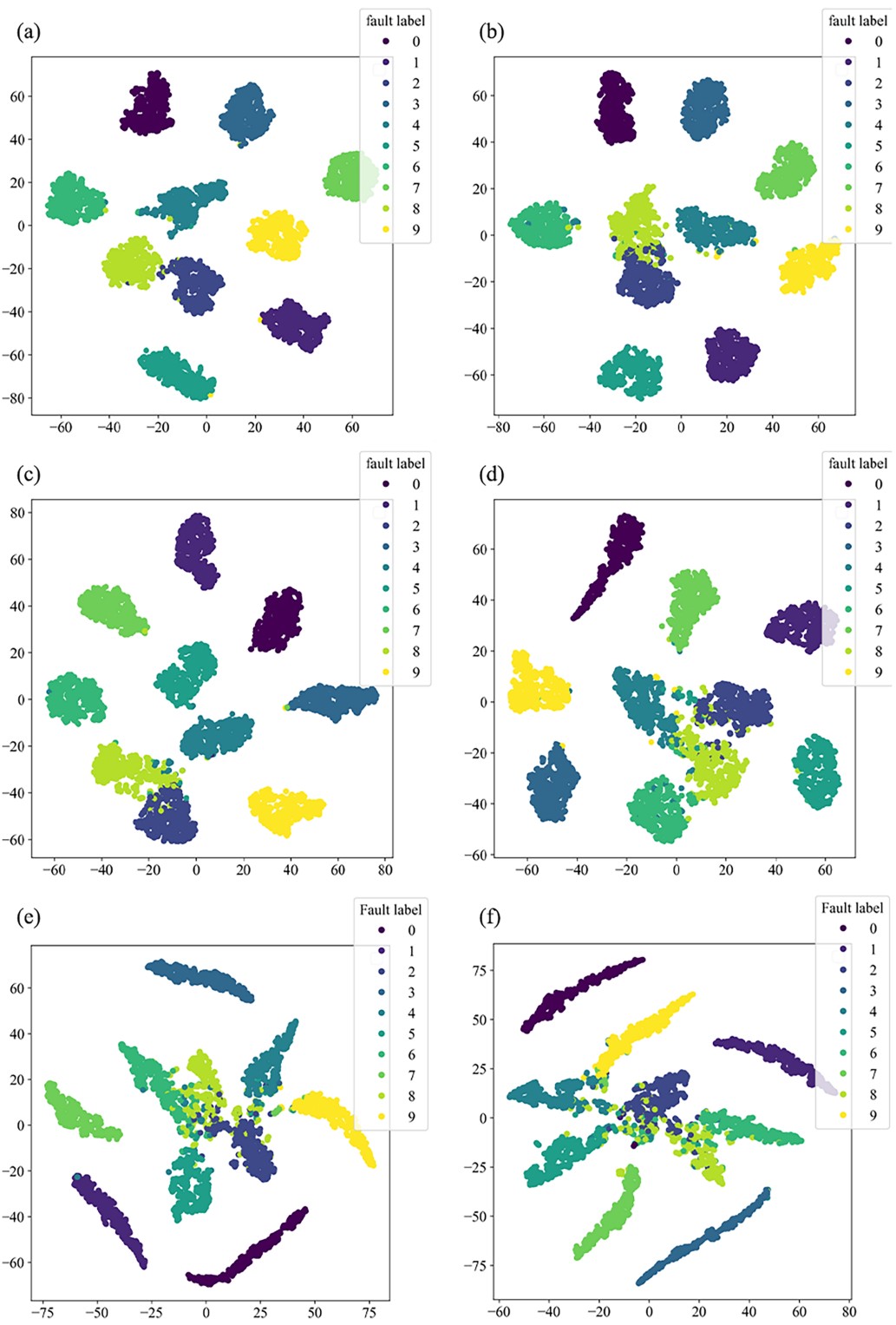

**Figure 9 t-SNE dimensionality reduction visualization.** (A) Clustering diagram of the fully connected layer for Fca-ResNet; (B) clustering diagram of the fully connected layer for SE-ResNet; (C) clustering diagram of the fully connected layer for ResNeXt; (D) clustering diagram of the fully connected layer for ResNet; (E) clustering diagram of the fully connected layer for GoogleNet; (F) clustering diagram of the fully connected layer for AlexNet.

**Table 4 Evaluation metrics for different network models.**

| Method | Accuracy % | Precision % (average) | Recall % (average) | F1 score |
|---|---|---|---|---|
| Proposed method | 99.3 | 99.28 | 99.25 | 99.26 |
| SE-ResNet | 98.8 | 98.78 | 98.75 | 98.76 |
| ResNeXt | 98.5 | 98.56 | 98.5 | 98.53 |
| ResNet | 98.5 | 98.63 | 98.5 | 98.56 |
| GoogleNet | 98 | 98.05 | 98 | 98.02 |
| AlexNet | 96.5 | 96.63 | 96.5 | 96.6 |
| LSTM-1DCNN | 98.46 | 98.21 | 98.45 | 98.06 |
| MTF-ResNet | 98.52 | 98.27 | 98.56 | 97.7 |
| ResNet-LSTM | 99.1 | 99.02 | 98.98 | 99.10 |

reduction effect is notably inferior to that of Fca-ResNet. This suggests that Fca-ResNet possesses very powerful feature extraction and classification capabilities.

## Comparison of different diagnostic algorithms

In order to demonstrate the effectiveness of the proposed method in this article, a comparison was conducted with other methods, using the highest accuracy as a benchmark for validation. *Chen, Zhang & Gao (2021)* directly input vibration signals into 1D-CNN and LSTM for training and classification; *Yan, Kan & Luo (2022)* transformed the vibration signals into two-dimensional images using the Markov transition field (MTF), and then utilized ResNet for training and classification; *Wang & Cheng (2021)* combined ResNet with LSTM to enable ResNet to capture the long-term correlations in time series, and achieved better classification performance.

In the field of deep learning fault diagnosis, the evaluation of models is very important. Only by selecting an evaluation method that matches, can problems in the algorithm model or training process be quickly discovered. Standard measures such as accuracy (*Ac*), precision (*Pr*), recall (*Re*), and F1 score (Eqs. (11)–(14)) are widely employed to evaluate model performance. The comparative results are as shown in Table 4.

$$Ac = \frac{TP + TN}{TP + FP + TN + FN} \tag{11}$$

$$Pr = \frac{TP}{TP + FP} \tag{12}$$

$$Re = \frac{TP}{TP + FN} \tag{13}$$

$$F1 = \frac{2Pr \times Re}{Pr + Re} \tag{14}$$

The above equation defines *TP* (true positives) and *TN* (true negatives) as the counts of correct predictions within the *i* categories, whereas *FP* (false positives) and *FN* (false negatives) represent the counts of incorrect predictions within the same *i* categories.

**Table 5 Imbalanced data sample division.**

| Fault type | Fault diameter (in) | Label | Datasets A1 | Datasets A2 | Datasets A3 |
|---|---|---|---|---|---|
| Inner ring failure | 0.007 | 1 | 100 | 300 | 200 |
| Inner ring failure | 0.014 | 2 | 100 | 300 | 200 |
| Inner ring failure | 0.021 | 3 | 100 | 300 | 200 |
| Rolling element failure | 0.007 | 4 | 200 | 100 | 300 |
| Rolling element failure | 0.014 | 5 | 200 | 100 | 300 |
| Rolling element failure | 0.021 | 6 | 200 | 100 | 300 |
| Outer ring failure | 0.007 | 7 | 300 | 200 | 100 |
| Outer ring failure | 0.014 | 8 | 300 | 200 | 100 |
| Outer ring failure | 0.021 | 9 | 300 | 200 | 100 |
| Normal | 0 | 0 | 400 | 400 | 400 |

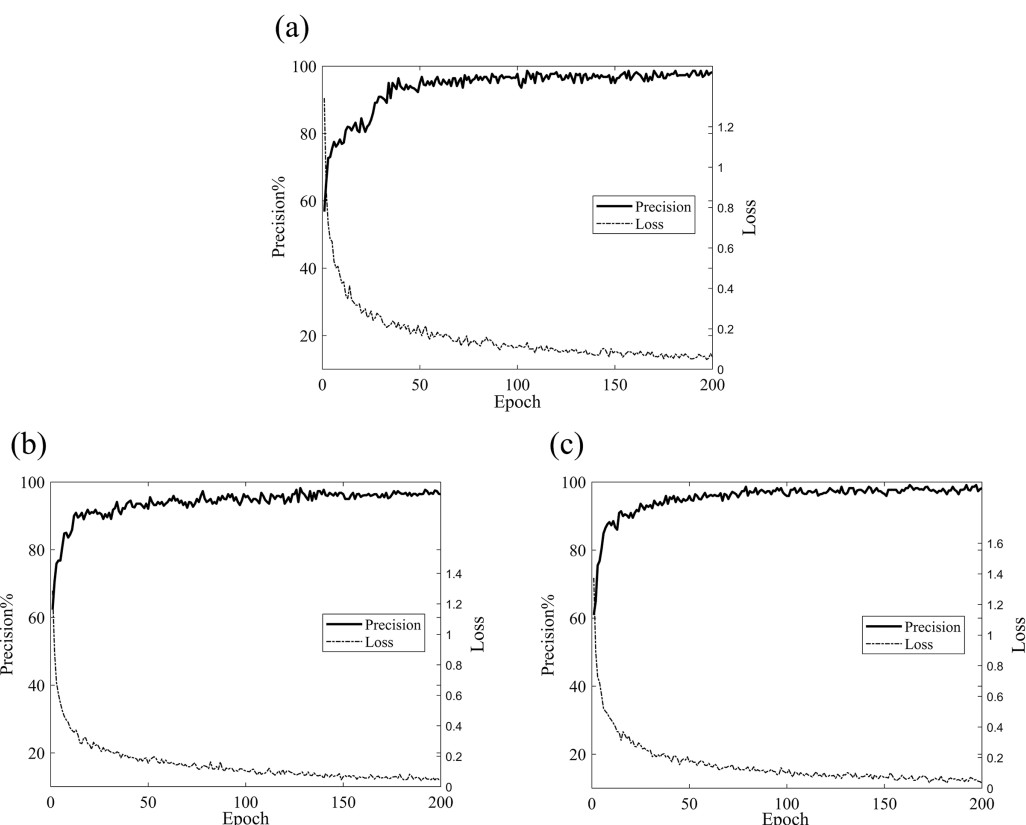

**Figure 10 Validation accuracy and loss curves for different datasets.** (A) Validation accuracy and loss curves for the A1 dataset; (B) validation accuracy and loss curves for the A2 dataset; (C) validation accuracy and loss curves for the A3 dataset.

## Verification of model generalization under imbalanced data sets

In this section, we continue to consider the case of 1 hp with a speed of 1,772 r/min. Taking into account the imbalance in the dataset that can occur in practical conditions, we divide

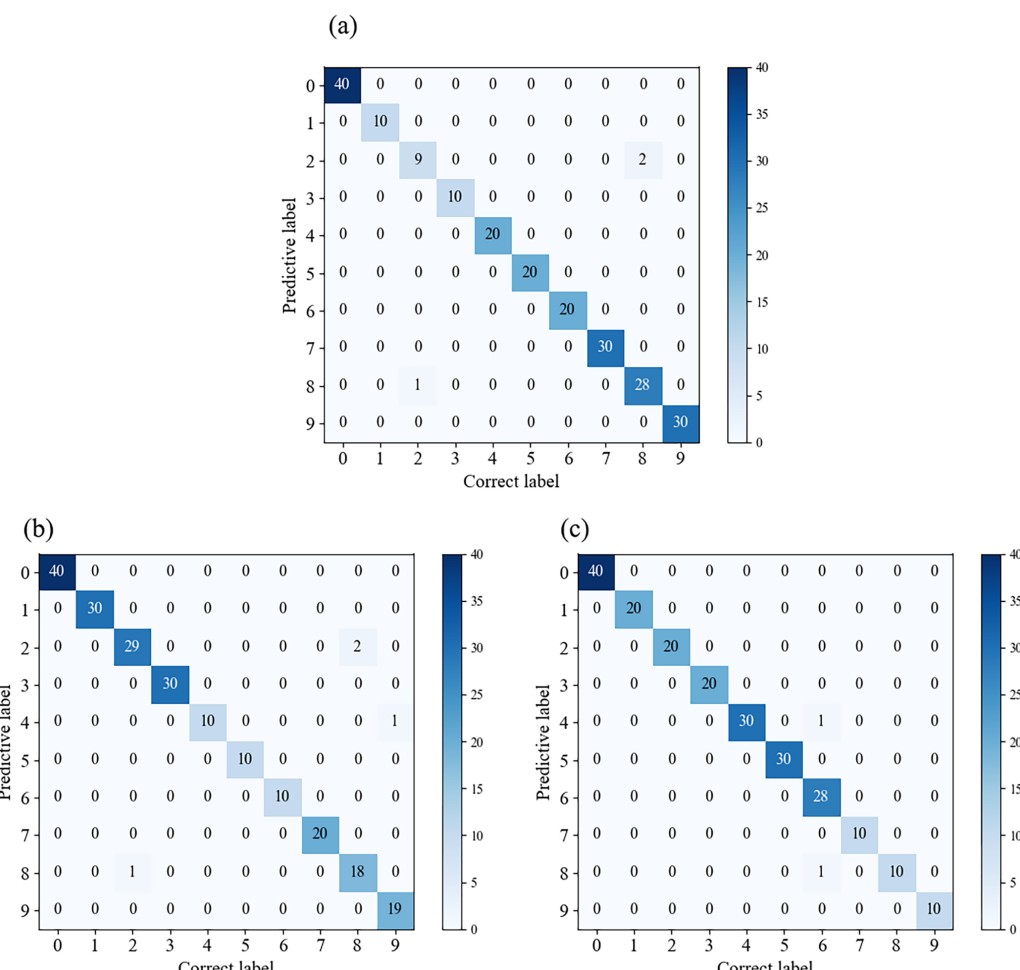

**Figure 11 Confusion matrices for different datasets.** (A) Confusion matrix for the A1 dataset; (B) confusion matrix for the A2 dataset; (C) confusion matrix for the A3 dataset.

the imbalanced datasets in different proportions. Datasets A1, A2, and A3 all simulate scenarios where there are adequate healthy samples and imbalanced faulty samples in actual working conditions. Imbalanced data samples are divided into training and validation sets, according to the ratio of 9:1, the division results are shown in Table 5.

Set the same training parameters according to Table 3, and set the number of training epochs to 200 for data validation under unbalanced datasets. The validation accuracy and loss curves are shown in Fig. 10. From the figure, it can be observed that, across different imbalanced datasets, the model's validation accuracy and loss tend to converge after approximately 30–50 training epochs. The highest validation accuracies are as follows: A1: 98.6%, A2: 98.2%, A3: 99.1%, with an average validation accuracy of 98.6%. The lowest training loss is consistently 0.03, achieving convergence effect, which demonstrates that this model exhibits a certain level of stability when dealing with imbalanced datasets.

Figure 11 displays the confusion matrices for different datasets. It can be observed that there are some misclassifications in individual categories for each class, while the rest

achieve full classification. Therefore, this model exhibits good generalization performance under various data conditions.

## CONCLUSION

(1) Our research employs the GADF coding technique for data preprocessing, which innovatively transforms one-dimensional fault data into a two-dimensional RGB image with three-channel encoding. This approach not only preserves the temporal characteristics of fault signals but also bolsters feature extraction and recognition in concert with the frequency channel attention module.

(2) Embedding the frequency channel attention module with discrete cosine transform on the basis of residual network to construct the Fca-ResNet model. This sophisticated model excels at extracting fault information, significantly enhancing the precision of fault diagnosis. The model's robustness is demonstrated by its remarkable 99.3% accuracy rate in diagnosing 1 hp load bearing faults using the rolling bearing data from CWRU.

(3) Comparative analysis with other models underlines the superior accuracy and swift convergence of the Fca-ResNet model. Through analysis methods such as t-SNE dimensionality reduction, confusion matrix, and model evaluation indicators, the strong classification performance of the model has been proven.

(4) In simulating several situations of unbalanced data sets in actual working conditions, the average accuracy of this model is 98.6%, which shows that this model still has good generalization in unbalanced samples.

## PROSPECTS

Future research in rolling bearing fault diagnosis will focus on improving the interpretability of deep learning models and combining multiple techniques to enhance model performance and reliability for better application of rolling bearing fault diagnosis in industrial production practice.

(1) To direct attention towards developing more interpretable deep learning models, such as using gradient or activation visualization methods to explain the model's decision-making process. Such techniques can help engineers understand the key features and rationale of the model in the diagnostic process, to enhance trust and reliability.

(2) To leverage transfer learning and incremental learning to improve the model's generalization ability, enabling it to better adapt to different types and scales of rolling bearing fault data, so as to enhance the model's robustness and reliability.

(3) To employ self-supervised learning and weakly supervised learning methods to reduce reliance on a large amount of labeled data, improve the model's performance in data-scarce scenarios, and enhance the model's interpretability.

(4) To establish a comprehensive model trustworthiness measurement system, including metrics and methods for model performance evaluation, error analysis, and model robustness assessment, to help engineers better understand and assess the reliability and applicability of the model in rolling bearing fault diagnosis.

### Funding

This work was sponsored by the Marine Power Research & Development (KY10300210082). The funders had no role in study design, data collection and analysis, decision to publish, or preparation of the manuscript.

### Grant Disclosures

The following grant information was disclosed by the authors:
Marine Power Research & Development: KY10300210082.

### Competing Interests

The authors declare that they have no competing interests.

### Author Contributions

- Lunpan Wei conceived and designed the experiments, performed the experiments, performed the computation work, prepared figures and/or tables, authored or reviewed drafts of the article, and approved the final draft.
- Xiuyan Peng conceived and designed the experiments, performed the experiments, prepared figures and/or tables, authored or reviewed drafts of the article, and approved the final draft.
- Yunpeng Cao analyzed the data, performed the computation work, prepared figures and/or tables, authored or reviewed drafts of the article, and approved the final draft.

### Data Availability

The data is available at GitHub and Zenodo:
- https://github.com/qwerty123456000000/data.git.
- qwerty123456000000. (2023). qwerty123456000000/data: data [Data set]. Zenodo. https://doi.org/10.5281/zenodo.10416722.

### Supplemental Information

Supplemental information for this article can be found online at http://dx.doi.org/10.7717/peerj-cs.1807#supplemental-information.

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
