# Peer review of "Rolling bearing fault diagnosis based on Gramian angular difference field and improved channel attention model"

_PeerJ Computer Science, doi:10.7717/peerj-cs.1807_

## Round 0.1 · original submission · Major Revisions

Based on the recommendations from the two reviewers, the paper needs a major revision.

Reviewer 1 ·

Basic reporting

The novelty of the proposed scheme is unclear, and the language needs substantial improvement.

Experimental design

The experimental design is too plain and not interesting enough.

Validity of the findings

This study does not appear to have any valuable findings. If possible, multidimensional comparative studies are needed. In addition, analysis of corresponding results should be supplemented.

Additional comments

1)The language of this manuscript needs considerable improvement. Currently, the language is extremely poor.
2)Lots of typos are observed, such as in Eqs. 1-3.
3) A large number of comparative experiments need to be supplemented. The current experimental results cannot reflect the advantages of the proposed scheme.
4)The format and dpi of all figures in this manuscript deserve improvement.

·

Basic reporting

The authors propose an enhanced fault diagnosis method for rolling bearing faults by combining Gramian Angular Difference Field (GADF) with residual networks (ResNet). To improve accuracy, the method incorporates channel attention modules within the ResNet architecture. Industrial bearing fault diagnosis remains a hot topic; however, this paper still has some issues. Therefore, my suggestion for this paper is a major revision.

Experimental design

no comment'

Validity of the findings

no comment'

Additional comments

(1) In Section 2, the authors mention that GADF retains temporal information. However, it is not clear how this is achieved. The statement "In order to retain temporal information, the values corresponding to time are distributed at certain angles on the span circle" lacks clarity in explaining how the temporal information is preserved. It would be helpful to provide a clearer explanation, preferably accompanied by Figure 1, to illustrate how the temporal information is preserved.

(2) Regarding the proposed FcaNet channel attention model in this paper, it appears that the authors have not provided the model's loss function or details about the training and optimization process.

(3) I am curious about the duration of the time span corresponding to the 300 data points in a window during the data preprocessing process. Additionally, for the accumulated bearing fault data, what is the fault frequency for each type of bearing fault data? I believe that the authors should provide explanations or presentations regarding this information.

(4) This paper lacks sufficient ablation experiments to demonstrate the superiority of the proposed model's modules and introduction methods.

(5) In Table 3 and Table 4, I believe that the number of compared state-of-the-art models is not enough, so the conclusions drawn are not very convincing. Also, why are only accuracy metrics displayed in the comparative results in Table 4? What was the purpose of displaying other metrics in Table 3, and are there any special insights?

(6) Why is there an empty page for the presentation of Fig. 2 and Tables 1-3? Furthermore, the legend positions in Fig.9 and Fig.10 need to be adjusted to avoid overlapping with the figures.

(7) I believe that providing future research prospects is necessary. For instance, interpretability has always been a challenge for the widespread application of deep learning in industrial fault diagnosis. The authors can base their future research prospects on this point. In fact, there have been related studies on offshore wind turbines that have provided some interpretability to models through the method of priors fusion, which the authors can refer to appropriately.

---

## Round 0.2 · Minor Revisions

Based on two review reports, the paper needs a minor correction.

Reviewer 1 has requested that you cite specific references. You may add them if you believe they are especially relevant. However, I do not expect you to include these citations, and if you do not include them, this will not influence my decision.

Reviewer 1 ·

Basic reporting

This revised manuscript is much improved.

Experimental design

A complete experimental result is presented. If possible, comparisons of more advanced methods need to be introduced.

Validity of the findings

The research findings match the experimental results.

Additional comments

1. Some typos are observed. For example, in abstract, “on the foundation of the ResNet=we embedded …” should be “on the foundation of the ResNet, we embedded …”. Please double check the entire manuscript.

2. Some new but important works in fault diagnosis should be discussed. Such as, [1] S. Lu, Z. Gao, Q. Xu, C. Jiang, A. Zhang and X. Wang, "Class-Imbalance Privacy-Preserving Federated Learning for Decentralized Fault Diagnosis With Biometric Authentication," IEEE Transactions on Industrial Informatics, vol. 18, no. 12, pp. 9101-9111, 2022. [2] Q. Xu, S. Lu, W. Jia, and C. Jiang, "Imbalanced fault diagnosis of rotating machinery via multi-domain feature extraction and cost-sensitive learning," Journal of Intelligent Manufacturing, vol. 31, no. 6, pp. 1467-1481, 2020.

3. Some abbreviations are not defined (FCA).

4. Labels 0 to 10 indicate what faults need to be clarified.

·

Basic reporting

I believe the authors have made targeted revisions based on my editing suggestions, and they have also addressed my related questions. I think this paper meets the publication requirements of the journal. I hope the authors will check for grammar and spelling issues before formal acceptance.

Experimental design

N/A

Validity of the findings

N/A

Additional comments

I believe the authors have made targeted revisions based on my editing suggestions, and they have also addressed my related questions. I think this paper meets the publication requirements of the journal. I hope the authors will check for grammar and spelling issues before formal acceptance.

---

## Round 0.3 · accepted · Accept

The paper is ready to be accepted.